# 2D perovskite stabilized phase-pure formamidinium perovskite solar cells

Jin-Wook Lee [1], Zhenghong Dai[1], Tae-Hee Han[1], Chungseok Choi[1], Sheng-Yung Chang[1], Sung-Joon Lee [1], Nicholas De Marco[1], Hongxiang Zhao [1], Pengyu Sun[1], Yu Huang [1] & Yang Yang[1]

Compositional engineering has been used to overcome difficulties in fabricating high-quality phase-pure formamidinium perovskite films together with its ambient instability. However, this comes alongside an undesirable increase in bandgap that sacrifices the device photo-current. Here we report the fabrication of phase-pure formamidinium-lead tri-iodide perovskite films with excellent optoelectronic quality and stability. Incorporation of 1.67 mol% of 2D phenylethylammonium lead iodide into the precursor solution enables the formation of phase-pure formamidinium perovskite with an order of magnitude enhanced photoluminescence lifetime. The 2D perovskite spontaneously forms at grain boundaries to protect the formamidinium perovskite from moisture and suppress ion migration. A stabilized power conversion efficiency (PCE) of 20.64% (certified stabilized PCE of 19.77%) is achieved with a short-circuit current density exceeding $24\,\mathrm{mA\,cm^{-2}}$ and an open-circuit voltage of 1.130 V, corresponding to a loss-in-potential of 0.35 V, and significantly enhanced operational stability.

[1] Department of Materials Science and Engineering and California NanoSystems Institute, University of California, Los Angeles, CA 90095, USA. Correspondence and requests for materials should be addressed to Y.Y. (email: yangy@ucla.edu)

Tremendous attention has been focused on hybrid perovskites (PVSK) since the first development of the solid-state PVSK solar cell in 2012[1–4]. Rapid progress in power conversion efficiency (PCE) has been achieved via compositional and process engineering. As of 2017, the state-of-the-art PVSK solar cell achieved a certified PCE of 22.7%, which is on par with well-established silicon solar cells[5–12].

Typical PVSK absorbers employ 3D $ABX_3$ structures, where a monovalent 'A-site' cation in the cubo-octahedral site bonds with the $BX_6$ octahedra. Compositional engineering has been considered an important approach to enhance the stability and performance of PVSK solar cells. Important milestones have been achieved through compositional engineering. For example, incorporation of the formamidinium (FA) cation into the 'A-site' has enabled the formation of a cubic $FAPbI_3$ phase with a lower bandgap ($E_g$) of 1.48 eV, higher absorption coefficient and longer carrier diffusion lengths than methylammonium (MA)-based tetragonal $MAPbI_3$ ($E_g = 1.57$ eV)[6,7,13]. However, $FAPbI_3$ has poor ambient stability because its non-PVSK hexagonal phase is thermodynamically more favorable than the cubic phase at room temperature. Partial substitution of FA and I with MA and/or Br has enabled fabrication of phase-pure $FAPbI_3$ with improved performance and stability[8]. Recently, incorporation of smaller inorganic 'A' cations, such as Cs and Rb, has further improved the stability and PCE of the PVSK solar cells with the lowest open-circuit voltage ($V_{OC}$) deficit of 0.39 V[14,15]. As a result, typical high efficiency devices nowadays incorporate PVSK with FA, MA, Cs, Rb, and Br having relatively larger $E_g$ than 1.60 eV[15,16]. However, such compositional engineering has enhanced the $V_{OC}$ and stability at the expense of short-circuit current density ($J_{SC}$) due to increased $E_g$. Utilization of pure $FAPbI_3$ is desired in regards to its lower $E_g$, which is close to the optimum value for a single junction solar cell suggested by the detailed balance limit[17]. However, no efficient method has been developed so far to fabricate a high quality phase-pure $FAPbI_3$ film and device.

Recently, the manipulation of surface energy has been proposed as a means to stabilize metastable PVSK phases such as $CsPbI_3$ and $FAPbI_3$[18–20]. Swarnkar et al. reported ambient stable α-$CsPbI_3$ in the form of a colloidal quantum dot, in which the contribution of surface energy significantly increases due to the high surface-to-volume ratio[18]. Very recently, Fu et al. reported that the cubic $FAPbI_3$ phase can be stabilized by functionalizing the surface with large-sized organic molecules[19]. They demonstrated that the functionalized surface contributes to lower formation energy to stabilize the cubic $FAPbI_3$ phase at room

temperature. However, the steady-state PCE was as low as 14.5%. Several attempts have been made to utilize such approach, where impressive improvements in performance and stability have been demonstrated[21–25]. However, their performance and stability are still relatively poor comparing with those of $MAPbI_3$ or mixed-cation-halide PVSK solar cells.

Here we report a method to fabricate high-quality stable $FAPbI_3$ PVSK films using 2D PVSK. Incorporation of 2D phenylethylammonium lead iodide ($PEA_2PbI_4$) PVSK into precursor solution enables the formation of phase-pure $FAPbI_3$ films with a tenfold enhancement in photoluminescence (PL) lifetime. The 2D PVSK is spontaneously formed at the grain boundaries of $FAPbI_3$ to protects the $FAPbI_3$ from moisture and assists in charge separation/collection. Thanks to the superior optoelectronic quality, we were able to fabricate a PVSK solar cell with a stabilized efficiency of 20.64% (certified stabilized efficiency of 19.77%). Notably, the PVSK solar cell shows a peak $V_{OC}$ of 1.130 V, corresponding to a loss-in-potential of 0.35 V considering the $E_g$ of 1.48 eV versus 0.39 V for mixed-cation-halide perovskite solar cells[15]. Furthermore, the device demonstrates significantly enhanced ambient and operational stability.

## Results

**Effects of 2D perovskite on phase purity of $FAPbI_3$.** $FAPbI_3$ films were prepared by the modified adduct method, in which N-methyl-2-pyrrolidone (NMP) was used as a Lewis base[26,27]. To the PVSK precursor solution, 2D PVSK ($PEA_2PbI_4$) precursors with different molar ratios ranging from 1.25 to 10 mol% were added. The steady-state PL spectra of the films were measured and are shown in Supplementary Fig. 1. As seen in Fig. 1a and Supplementary Fig. 1, we observed no obvious changes in PL peak position until the amount of 2D PVSK reached 10 mol%. With 10 mol% PVSK, the PL peak was blue-shifted by 6 nm. The blueshift of the PL peak might be due to formation of a quasi-3D PVSK, where charge carriers are confined by large potential barrier originated from the 2D PVSK[28]. Based on this observation, we presume the added 2D PVSK does not result in the formation of the quasi-3D PVSK if it remains below a certain threshold. This threshold was found to be lower than 10 mol%, where this quantity was then optimized based on photovoltaic performance (Fig. 1a and Supplementary Fig. 2). A planar heterojunction architecture consisting of Indium doped $SnO_2$ (ITO) glass/compact-$SnO_2$/PVSK/spiro-MeOTAD/Ag or Au was utilized for construction of PVSK solar cells in this study (cross-sectional scanning electron microscopic (SEM) images of the

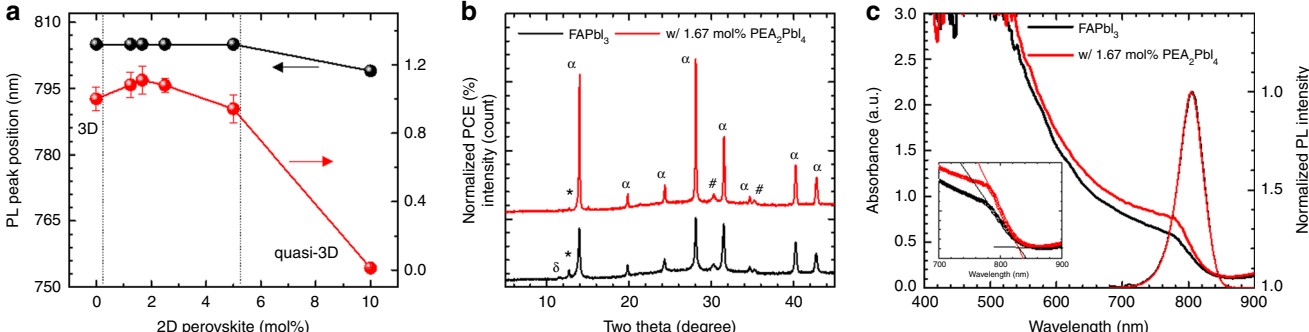

**Fig. 1** Crystallographic and absorption properties. **a** Peak position for steady-state photolumimescence (PL) spectrum and normalized power conversion efficiency (PCE) of the devices for $FAPbI_3$ perovskite with different amount of added 2D $PEA_2PbI_4$ perovskite. The error bar of the normalized PCE indicates standard deviation of the PCEs. At least 10 devices were fabricated for each condition. **b** X-ray diffraction patterns, **c** absorption and normalized PL spectra of bare $FAPbI_3$ and $FAPbI_3$ with 1.67 mol% $PEA_2PbI_4$ 2D perovskite. Inset of **c** shows onset region of the absorption spectra with linear approximation (solid lines)

device are shown in Supplementary Fig. 3). The addition of 1.67 mol% 2D PVSK was found to be optimal for the photovoltaic performance (ca. 11% improvement in PCE). Notably, addition of 10 mol% 2D PVSK significantly degraded the PCE to lower than 1%, which might result from formation of quasi-3D PVSK as the large potential barrier originating from 2D PVSK could hinder the charge transport.

X-ray diffraction patterns (XRD) of bare $FAPbI_3$ and $FAPbI_3$ with 1.67 mol% $PEA_2PbI_4$ are shown in Fig. 1b. As can be seen, the bare $FAPbI_3$ film contains hexagonal non-PVSK phase (δ-phase) while the PVSK film prepared with 1.67 mol% $PEA_2PbI_4$ shows pure PVSK phase[29]. Even smaller amount of 2D PVSK (1.25 mol% $PEA_2PbI_4$) effectively suppresses the formation of δ-phase (Supplementary Fig. 4). Furthermore, the overall signal intensity and full-width-half-maximum (FWHM) were enhanced with the addition of the 2D PVSK, indicating improved crystallinity. We speculate that the added large phenylethylammonium molecules from 2D PVSK precursors interact with $FAPbI_3$ crystals to facilitate formation of the cubic PVSK phase during crystallization[19]. Such a speculation is correlated with the observation in the XRD measurements in Supplementary Fig. 4, in which the signal intensity and FWHM of XRD peaks are systematically enhanced with increased amounts of the added 2D PVSK (Supplementary Figs. 4, 5). The enhancement of preferred orientation along the (001) plane with increased 2D PVSK also indicates the added precursors of the 2D PVSK functionalize the specific crystal facet to change the surface energy during the crystal growth[30]. A closer inspection on the normalized X-ray diffraction (XRD) patterns of the PVSK films with different amounts of added 2D PVSK (Supplementary Fig. 6) was taken to find any correlations between the added 2D PVSK and crystal structure of $FAPbI_3$. Interestingly, a systematic change in peak position was observed with different amounts of 2D PVSK for

which the XRD peaks were slightly shifted toward higher angles with the addition of relatively smaller amounts of 2D PVSK (1.25, 1.67, 2.50, and 5.0 mol%). This indicates that the lattice constant of $FAPbI_3$ is reduced, likely due to compressive strain associated with the added 2D PVSK. We speculate that the reduction in lattice constant can be also related to the enhanced phase purity of cubic $FAPbI_3$ as it will have equivalent effects with incorporation of smaller 'A' site cations on the tolerance factor and thus enthalpy of formation[31]. Lower angle peaks at around 12° appear upon addition of 10 mol% 2D PVSK corresponding to the formation of quasi-3D PVSK (inset of Supplementary Fig. 4f)[32]. The pure phase PVSK film with 1.67 mol% $PEA_2PbI_4$ shows enhanced absorption over all wavelengths (Fig. 1c) compared to the bare $FAPbI_3$ film where the absorption onset is hardly changed (Insent of Fig. 1c). The absorption onset is complemented by almost identical normalized PL spectra, which indicates that the $E_g$ was maintained. The enhanced absorption as seen when the 2D PVSK was added is probably due to an enhanced phase purity of the $FAPbI_3$, with partial contribution from an enhanced light scattering owing to the improved crystallinity[33]. The absorption spectra with different amounts of 2D PVSK are demonstrated in Supplementary Fig. 7. While all the PVSK films with 2D PVSK showed enhanced absorption compared to bare $FAPbI_3$ films, a slight blueshift of the absorption onset with decreases in absorption over the whole-wavelength region was identified with the addition of 10 mol% of 2D PVSK, which is correlated with the blueshift of the steady-state PL spectrum that can be associated with the formation of quasi-3D PVSK.

**Photoluminescence properties and photovoltaic performance.** Steady-state and time-resolved PL profiles were investigated in

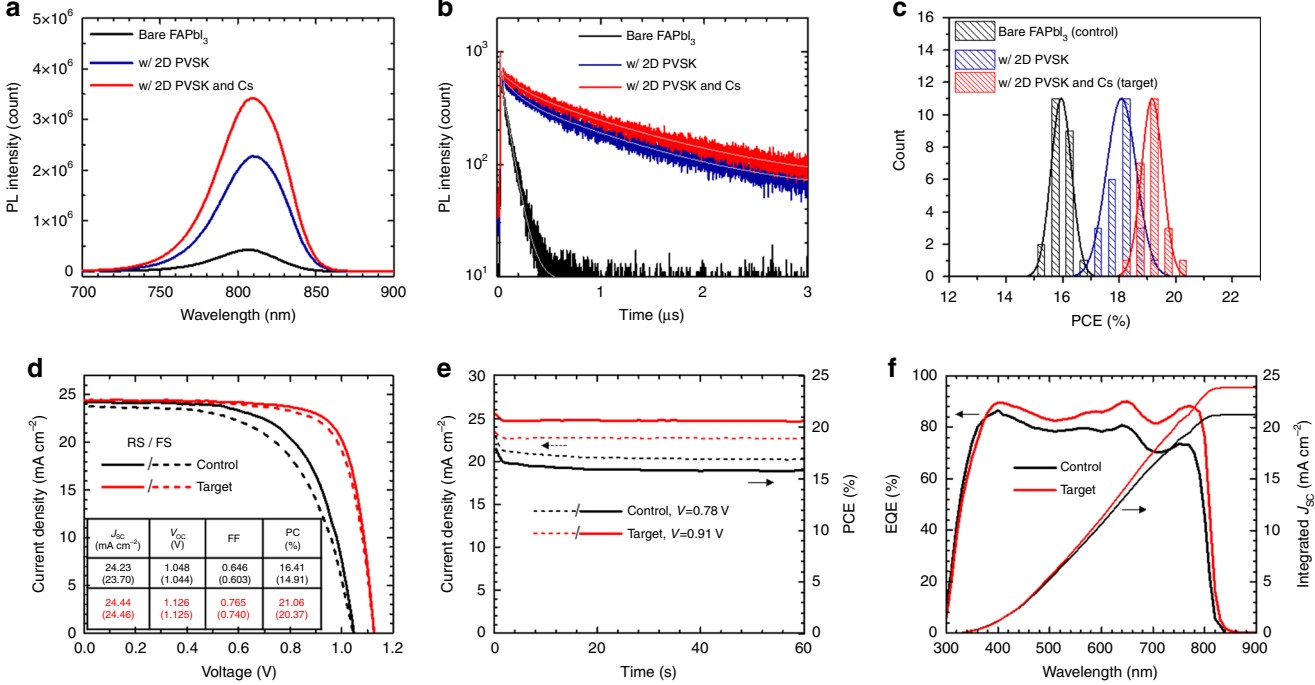

**Fig. 2** Photoluminescence properties and photovoltaic performance. **a** Steady-state and **b** time resolved PL spectra of the perovskite films incorporating bare $FAPbI_3$, $FAPbI_3$ with 2D perovskite and $FA_{0.98}Cs_{0.02}PbI_3$ with 2D perovskite. Gray solid lines in **b** are fitted lines for each curve. **c** Power conversion efficiency (PCE) distribution of the devices incorporating the perovskites. All the devices were fabricated in same batch. **d** Current density–voltage (J–V) curves, **e** steady-state PCE measurement and **f** external quantum efficiency (EQE) spectra of perovskite solar cells incorporating bare $FAPbI_3$ (control) and $FA_{0.98}Cs_{0.02}PbI_3$ with 2D perovskite (target). Photovoltaic parameters of the highest performing devices are summarized in the table in **d**, in which the values with and without parenthesis are from reverse (from $V_{OC}$ to $J_{SC}$) and forward scan (from $J_{SC}$ to $V_{OC}$), respectively

Fig. 2a, b. The steady-state PL intensity was largely enhanced more than five times from $4.3 \times 10^5$ to $2.3 \times 10^6$ with addition of 1.67 mol% $PEA_2PbI_4$ into $FAPbI_3$ film (Fig. 2a). The large enhancement of PL intensity was attributed to a significantly elongated PL lifetime as seen in Fig. 2b. The time resolved PL profiles were fitted to exponential decay, in which bi- and tri-exponential decay models were used for the bare and 2D PVSK incorporated PVSK films, respectively (Supplementary Table 1). The relatively fast decay component ($\tau_1$ around 3 ns) was assigned to charge carrier trapping induced by trap states formed due to the structural disorder such as vacancy or interstitial defects while much slower components ($\tau_2$, $\tau_3$) were assigned to free carrier radiative recombination[34–37]. With addition of 1.67 mol% 2D PVSK, proportion of the fast decay component ($\tau_1$) was decreased (from 51.8% to 46.5%) while $\tau_2$ significantly elongated from 78.5 ns to 148.7 ns, which indicates reduced defect density and enhanced charge carrier lifetime. We attributed such improvements to enhanced phase purity and crystallinity of $FAPbI_3$ as observed from XRD measurements (Fig. 1b), which decreases the structural disorders at grain interiors and/or boundaries[38]. Moreover, a new decay component ($\tau_3$) with a significantly long lifetime (>1 μs) appeared after addition of the 2D PVSK, which is likely related to the added 2D PVSK. As a result, the average PL lifetime was enhanced by almost one order of magnitude from 39.4 to 376.9 ns with addition of 2D PVSK. During the optimization of the device, incorporation of 2 mol% Cs was found to further enhance the performance and reproducibility of the devices without a noticeable change in $E_g$ (Supplementary Figs. 8–12, see Supplementary Notes 1, 2 and 3 for additional discussion on the optimization process and impacts of 2 mol% Cs). With additional 2 mol% Cs, the fraction of $\tau_1$ was further decreased, indicating a further decreased defect density, which was also observed in previous studies[14]. Consequently, the steady-state PL intensity and average PL lifetime was further enhanced, rationalizing the improved PCE with 2 mol% Cs (Supplementary Table 1). It is worth noting that the PL lifetime was significantly reduced with 10 mol% of 2D PVSK due to formation of quasi-3D PVSK (Supplementary Fig. 13).

The PCE distribution of the devices incorporating corresponding the PVSKs is compared in Fig. 2c (distribution of photovoltaic parameters can be found in Supplementary Fig. 14). The average photovoltaic parameters are summarized in Supplementary Table 2. The average PCE of the bare $FAPbI_3$ PVSK solar cells was significantly enhanced by 13% from $15.95 \pm 0.36\%$ to $18.08 \pm 0.52\%$ with addition of 1.67 mol% $PEA_2PbI_4$. The average PCE was further enhanced to $19.16 \pm 0.37\%$ with 2 mol% of Cs (Hereafter, the devices based on bare $FAPbI_3$ are denoted as control while the devices based on $FA_{0.98}Cs_{0.02}PbI_3$ with 1.67 mol % $PEA_2PbI_4$ are denoted as target for convenience). Current density and voltage (J–V) curves of the optimized control and target devices are demonstrated in Fig. 1d, in which the highest PCE of the target device reached 21.06% ($J_{SC}$: 24.44 mA cm$^{-2}$, $V_{OC}$: 1.126V, FF: 0.765) while a PCE of 16.41% was achieved with the control device ($J_{SC}$: 24.23 mA cm$^{-2}$, $V_{OC}$: 1.048V, FF: 0.646). A stabilized PCE of 20.64% was achieved with the target device while that of control device was 15.80% (Fig. 2e). External quantum efficiency (EQE) spectra of the devices were compared in Fig. 2f. An integrated $J_{SC}$ of 23.9 mA cm$^{-2}$ from the target device was well-matched with the value measured from the J–V scan (<5% discrepancy), while control device shows that of 21.2 mA cm$^{-2}$ with a relatively large discrepancy of 14%. The relatively large discrepancy from the control $FAPbI_3$ device is probably due to a more pronounced hysteresis, as seen in Fig. 1d, which also results in a large discrepancy between the stabilized PCE and the PCE measured from the J–V scan. The performance of control device was highly reproducible. With optimized

process parameters, the average PCE of $20.05 \pm 0.45\%$ was demonstrated over 74 devices (Supplementary Fig. 15). We obtained the peak $V_{OC}$ of 1.130 V with the target device (Supplementary Fig. 16) corresponding to a loss-in-potential of 0.35 V considering a $E_g$ of 1.48 eV, which is the lowest $V_{OC}$ deficit reported to date for PVSK solar cells. One of the target devices was sent out for measurement in an independent laboratory and achieved a certified stabilized PCE of 19.77% (Supplementary Fig. 17). The current–voltage curve and EQE spectra matched well with those measured by our group (Supplementary Fig. 18). The enhanced device performance with 2D PVSK is mainly due to improved FF and $V_{OC}$, which can be attributed to improved phase purity and elongated carrier lifetime with reduced defect density, facilitating carrier transport and reducing the charge recombination[14]. The reduced non-radiative recombination loss with 2D PVSK was also confirmed in devices by electroluminescence (EL) measurements in Supplementary Fig. 19, in which maximum radiance (40.4 Wsr$^{-1}$ cm$^{-2}$) and EL EQE (0.49%) of the target devices were significantly enhanced compared to those of the control devices (2.87 Wsr$^{-1}$ cm$^{-2}$, 0.06%)[15].

**Moisture stability and TEM analysis.** Under ambient conditions, a cubic $FAPbI_3$ PVSK phase is subject to undergo conversion to a hexagonal non-PVSK phase, resulting in serious degradation in photovoltaic performance[8,14]. The phase transformation is even accelerated under high-relative humidity[14]. To evaluate the effects of 2D PVSK incorporation on phase stability, we investigated changes in absorbance of the film under relative humidity (RH) of $80 \pm 5\%$. Figure 3a shows photos of the PVSK film stored for different time. Bare $FAPbI_3$ film was almost completely bleached within 24 h whereas no obvious change in color was observed from the films containing 2D PVSK both with and without Cs. Figure 3b demonstrates the absorbance (at 600 nm) of the $FAPbI_3$ films with and without 2D PVSK as a function of exposure time (individual absorption spectra can be found in Supplementary Fig. 20). The absorbance of the bare $FAPbI_3$ rapidly degraded during 24 h, while $FAPbI_3$ films with 2D PVSK did not show noticeable degradation within 24 h. With addition of 2 mol% Cs, the film also remained stable after 24 h. The color change of the bare $FAPbI_3$ film under high RH is due to its transformation to the δ-phase as can be seen in XRD spectra in Supplementary Fig. 21a whereas no detectable change in color for the films with 2D PVSK is correlated with their neat XRD spectra without the δ-phase (Supplementary Figs. 21b, c). The enhanced phase stability under high RH implies that the possible ingression pathway of moisture in the PVSK film is passivated. Previously, we demonstrated grain boundary engineering techniques using the adduct approach, in which the additives had precipitated at grain boundaries if not incorporated into the lattice of PVSK[34,38]. We supposed that grain boundaries within the film, which have been reported to be ingression pathways for moisture, might be passivated by the added 2D PVSK[39].

Indeed, the vertically aligned 2D PVSK was sparsely observed from SEM images in Supplementary Fig. 10b, c with addition of 2D PVSK (see also Supplementary Fig. 22 and Supplementary Note 4 for additional discussion). However, the enhanced moisture stability throughout the whole-film implies that the 2D PVSK probably exist along the grain boundaries. To confirm our assumption, transmission electron microscopic (TEM) images of the $FAPbI_3$ film with 2D PVSK was analyzed in Fig. 3c–e. The inset of Fig. 3c shows a chunk of the polycrystalline film scratched off from the substrate. Several hundreds of nanometer sized grains and their boundaries are clearly visible from the image, and from which one of the grains is magnified in Fig. 3c. The region (1) in Fig. 3c, which is the grain interior, was

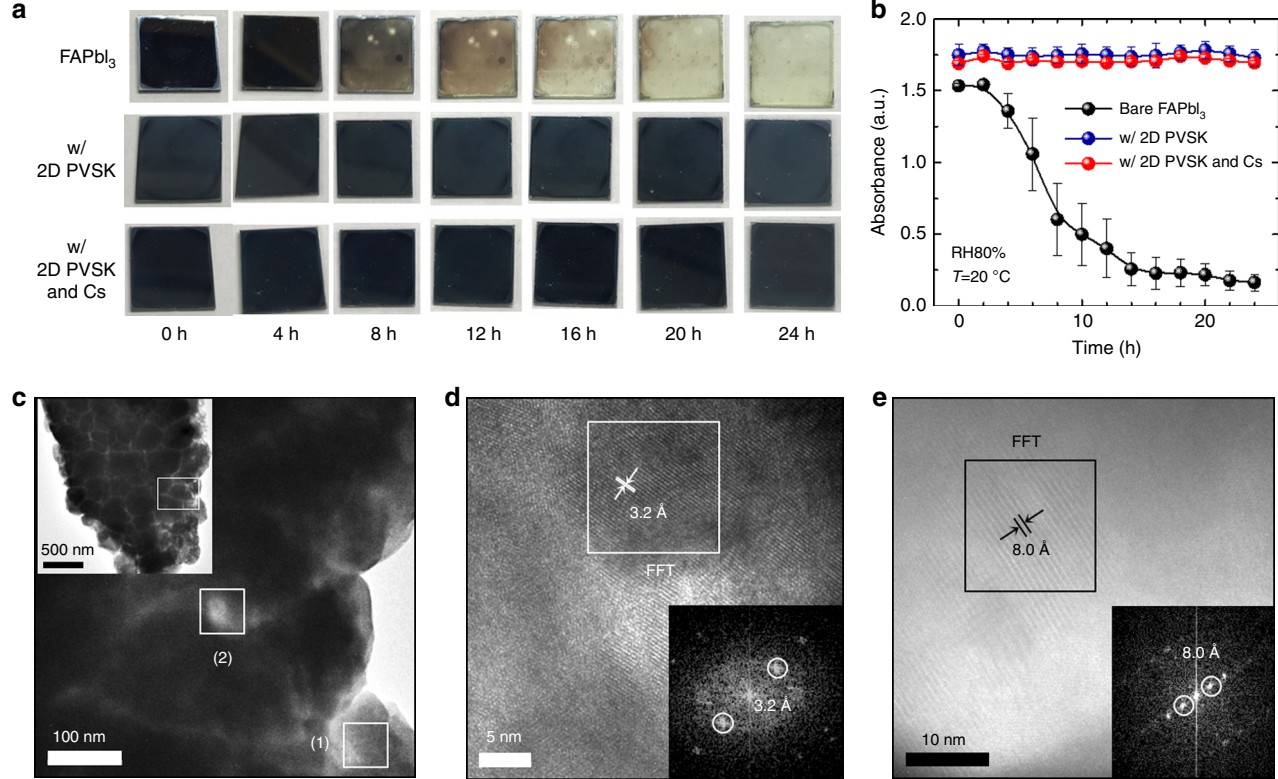

**Fig. 3** Improved moisture stability with 2D perovskite at grain boundaries. **a** Photos of the perovskite films incorporating bare FAPbI$_3$, FAPbI$_3$ with 2D perovskite and FA$_{0.98}$Cs$_{0.02}$PbI$_3$ with 2D perovskite exposed to relative humidity (RH) of 80 ± 5% at 20 ± 2 °C for different time. **b** Evolution of absorption of the films at 600 nm under RH 80 ± 5% at 20 ± 2 °C. The error bar indicates standard deviation of the absorbance measured from three films for each condition. **c–e** Transmission electron microscopic (TEM) images of the FA$_{0.98}$Cs$_{0.02}$PbI$_3$ film with 1.67 mol% PEA$_2$PbI$_4$. Inset of **c** demonstrates the lower magnification image showing the polycrystalline nature with grain boundaries. The highlighted area (1) and (2) were investigated in **d** and **e**, respectively. Inset of **d** and **e** show Fast Fourier transform (FFT) analysis of the area within boxes, respectively

magnified and analyzed using Fast Fourier transform (FFT) in Fig. 3d, in which an inter-planar spacing of 3.2 Å is well-matched with the (002) reflection of cubic FAPbI$_3$ (Supplementary Table 3). At region (2), which is grain boundary, the FFT analysis revealed an inter-planar distance of 8.0 Å (Fig. 3e), correlating to a characteristic (002) reflection of 2D PEA$_2$PbI$_4$ (Supplementary Fig. 23 and Supplementary Table 4). This supports the presence of 2D PVSK at grain boundaries, which was further confirmed by elemental distribution (EDS) analysis (Supplementary Fig. 24). At grain boundary regions, relatively larger amounts of carbon and nitrogen were detected, which could be due to presence of phenylethylammonium cation in the 2D PVSK.

**Band structure and electrical properties.** A schematic in Fig. 4a shows 2D PVSK formation at the grain boundaries of the 3D PVSK film. Since the 2D PEA$_2$PbI$_4$ PVSK with aromatic rings and longer alkyl chains is expected to be more resistant to moisture, it protects the defective grain boundaries of 3D PVSK, resulting in significantly enhanced moisture stability of the film. Regardless of the improved stability, however, one can expect degraded electronic properties of the film due to the poor charge carrier mobility of the 2D PVSK. We investigated the band structure of FA$_{0.98}$Cs$_{0.02}$PbI$_3$ (with 1.67 mol% of 2D PVSK) and PEA$_2$PbI$_4$ PVSK, which is illustrated in Fig. 4b. The valence band maximum was measured using ultraviolet photoelectron spectroscopy (UPS, Supplementary Fig. 25), while the $E_g$ was determined from Tauc plots (Supplementary Fig. 26). As seen in Fig. 4b, FA$_{0.98}$Cs$_{0.02}$PbI$_3$

and PEA$_2$PbI$_4$ PVSK shows type I band alignment. Such band alignment resembles the alignment between PVSK and PbI$_2$ formed at grain boundaries, which was found to reduce charge recombination and assist in charge separation/collection[40,41]. Thus, analogous advantages of 2D PVSK at grain boundaries can be expected. Conductive atomic force microscopy (c-AFM) was performed in Fig. 4c–f to see spatially resolved electrical properties of the films. Under ambient light conditions (Fig. 4c, d), current flow in the PVSK film with 2D PVSK was higher at/near the grain boundaries while relatively uniform current flow was observed in the bare FAPbI$_3$ film. With light illumination (Fig. 4e, f), the current flow was further enhanced at/near the grain boundaries with 2D PVSK whereas current flow in bare FAPbI$_3$ film was uniformly increased, which indicates charge separation and collection of photo-generated electrons is facilitated more so at grain boundaries with 2D PVSK. As suggested for PbI$_2$, thin 2D PVSK regions at grain boundaries might suffer downward band bending under illumination (dashed line in Fig. 4b) where photo-generated electrons are transferred from grain interiors. Due to the high-potential barrier to the holes, charge recombination will be reduced, which might be the origin of the superior PL lifetime and photovoltaic performance with 2D PVSK.

**Ambient and operational stability.** Finally, the stability of the control and target devices was compared. Figure 5a demonstrates the changes in PCE of the unencapsulated devices stored in a desiccator (relative humidity lower than 30%, evolution of an individual photovoltaic parameter can be found in

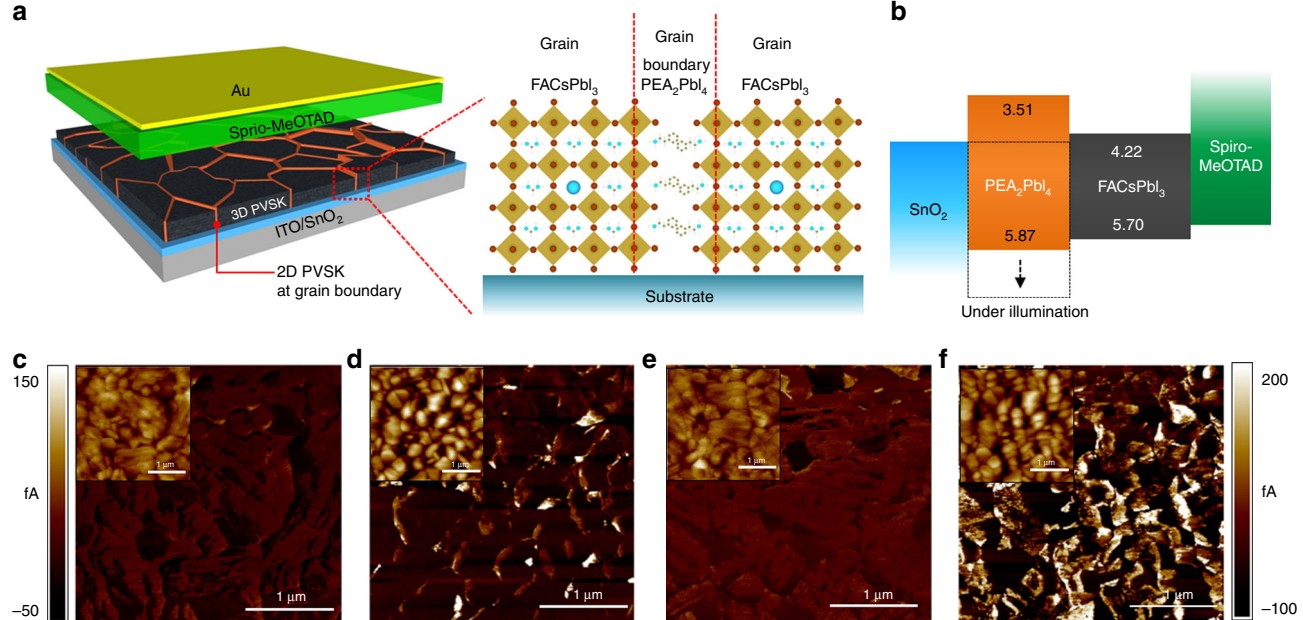

**Fig. 4** Band alignment and local conductivity with 2D perovskite. **a** Schematics of the device incorporating polycrystalline 3D perovskite film with 2D perovskite at grain boundaries and **b** band structure of each layer in device analyzed by ultraviolet photoelectron spectroscopy (UPS) and Tauc plots. Conductive atomic force microscopic (c-AFM) images of (**c**, **e**) bare FAPbI$_3$ and (**b**, **d**) with 2D perovskite films on SnO$_2$ coated ITO glass. The measurement was carried out with bias voltage of 100 mV under (**c**, **d**) room light or (**d**, **f**) low intensity light illumination provided by the AFM setup. Inset of each image shows corresponding topology of the films. Scale bar at left side is for (**c**) and (**d**) while at right side is for **e** and **f**

Supplementary Fig. 27). While the control device degraded by 29% for 1392 h, the target device maintained 98% of its initial efficiency during this time. The operational stability of the devices was also compared by maximum power point (MPP) tracking under 1 sun illumination in Fig. 5b. Without encapsulation, the PCE of the control device rapidly degraded by 68% during 450 min whereas that of target device was relatively less (20%) during the time. We performed 500 h of light exposure test with the encapsulated control (bare FAPbI$_3$ device) and target devices (w/ 1.67 mol% 2D PVSK). The encapsulated devices were exposed to ca. 0.9 sun (90 ± 5 mW cm$^{-2}$) under open-circuit condition, of which the steady-state PCE was periodically measured for different exposure time. As seen in Fig. 5c, both of the devices showed a rapid initial decay in PCE followed by slower decay with an almost linear profile, which is in agreement with previous reports[42]. After 500 h of exposure, the control device degraded to ca. 52.3% of its initial PCE whereas the target device maintained 72.3% of the initial PCE, indicating enhanced stability with addition of 2D PVSK. We could extract tentative T80 (time at which PCE of the device decays to 80% of initial PCE) for the devices by fitting of the post-burn-in region in which the PCE of the device shows an almost linear decay profile (after 48 h). The T80 for control and target devices were calculated to be 592 h and 1362 h, respectively. This indicates the stability of the device was significantly improved with addition of 2D PVSK. We also performed MPP tracking of the encapsulated target device under 1 sun (100 mW cm$^{-2}$) illumination in Supplementary Fig. 28. A total of 18.7% of initial PCE was degraded for 130 h of operation, which is relatively slower compared to the device maintained at open-circuit condition. This is correlated with previous studies that attributed the faster degradation under open-circuit condition to larger number of photo-generated charge carriers recombining within the device[43]. Under operational condition with abundant photo-generated charges and built-in electric field, the major factors causing the degradation of the devices might be the highly mobile and reactive charged defects (ions) and/or

trapped charge carriers associated with it[42,44]. We suppose that migration of the charged defects or ions is possibly suppressed by 2D PVSK at grain boundaries. The temperature-dependent conductivity ($\sigma$) measurement of the lateral devices was performed to evaluate the activation energy for the ion migration (Fig. 6). The activation energy ($E_a$) for the migration can be determined according to the Nernst-Einstein relation[42],

$$\sigma(T) = \frac{\sigma_0}{T} \exp\left(\frac{-E_a}{kT}\right),$$

where $k$ is Boltzmann constant, $\sigma_0$ is a constant. Inset of Fig. 6a describes the structure of the lateral devices. With bare FAPbI$_3$ PVSK, exponential enhancement in conductivity was clearly identified at around 130 K (Fig. 6a), which is attributed to contribution of ions. The $E_a$ for bare FAPbI$_3$ film was calculated to be 0.16 eV, indicating significant contribution of activated ions at room temperature, which might cause degradation of the material and device under operational condition with built-in electric field. The pronounced current–voltage hysteresis behavior was observed even at very low temperature (180 K, Fig. 6c). In case of the PVSK film with 2D PVSK, the film did not show noticeable enhancement in conductivity with increased temperature; although, the overall conductivity was relativity lower than the bare FAPbI$_3$ film (Fig. 6c). Even with moderate light illumination, it does not show the indicative of activated ions. As a result, the current–voltage curve did not show any hysteresis behavior (Fig. 6d). As the grain boundaries of 3D perovskite were reported to be a major pathway for the migration of ions[39], passivating the grain boundaries by incorporation of the ion-migration-immune 2D PVSK likely suppressed overall ion migration in the target device[45]. In addition, the improved phase purity of the film might also partially contribute to the suppressed ion migration because the secondary phase can generate defect sites that can act as an additional pathway for ion migration. We believe the suppressed

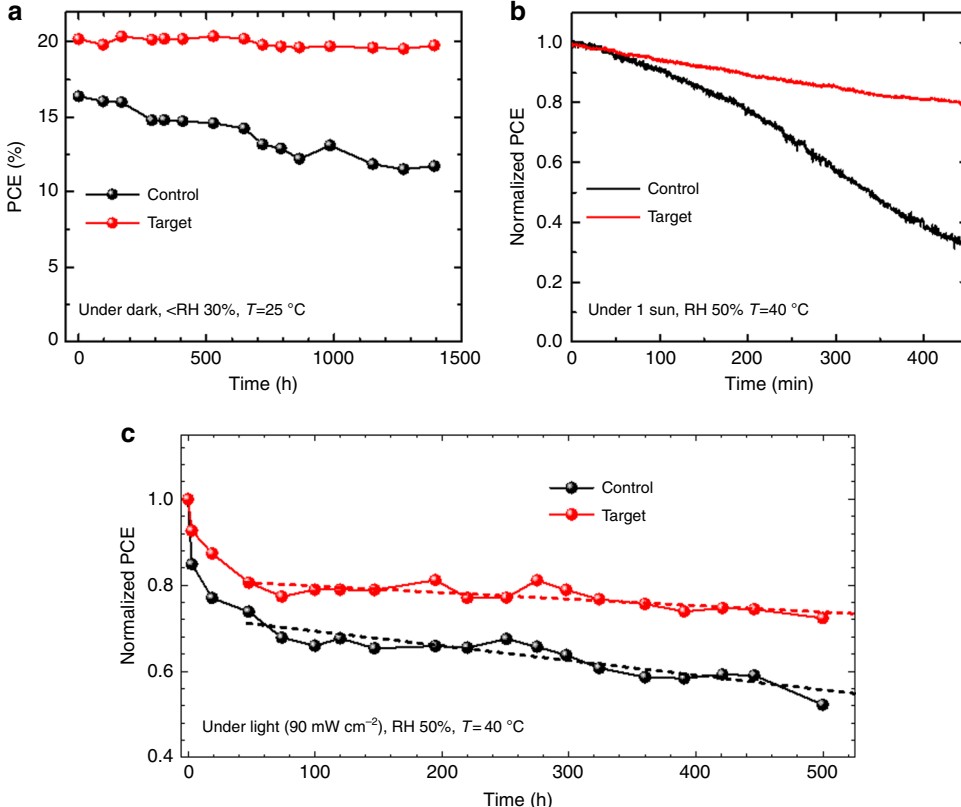

**Fig. 5** Improved stability with 2D perovskite. **a** Evolution of power conversion efficiency (PCE) of control and target devices. The devices were stored under dark with controlled humidity. **b** Maximum power point tracking of the devices under 1 sun illumination in ambient condition without encapsulation. **c** Evolution of the PCEs measured from the encapsulated control and target devices exposed to continuous light ($90 \pm 5$ mW cm$^{-2}$) under open-circuit condition. The stabilized PCEs were measured at each time. Initial stabilized PCEs for control and target devices were 14.5% and 17.5%, respectively. The broken lines are linear fit of the post-burn-in region (after 48 h). Relative humidity (RH) and temperature ($T$) are indicated in the graphs for each measurement

ion migration contributes to enhanced operational stability of the target device.

## Discussion

We demonstrated a reproducible way to fabricate phase-pure formamidinium tri-iodide PVSK with high-optoelectronic quality and stability by incorporating 2D PVSK. The large phenylethylammonium molecules from 2D PVSK precursors interact with FAPbI$_3$ crystals to facilitate formation of the cubic PVSK phase during crystallization, which subsequently functionalize the grain boundaries after completion of the crystallization. The resulting phase-pure PVSK film has an identical $E_g$ (1.48 eV) to that of pure FAPbI$_3$ with an order of magnitude enhanced PL lifetime. Average PCE of $20.05 \pm 0.45\%$ over 74 devices and the highest stabilized PCE of 20.64% (certified stabilized PCE of 19.77%) was achieved. Regardless of its low $E_g$, the PVSK solar cell showed the peak $V_{OC}$ of 1.130 V, corresponding to the lowest loss-in-potential of 0.35 V among all the reported PVSK solar cells. Owed to the functionalized grain boundaries by the 2D PVSK, the phase stability of the film under high RH significantly improved and migration of ions (or charged defects) was suppressed, resulting in significantly improved ambient and operational stability of the device. We believe our approach to utilize spontaneously formed grain boundary 2D PVSK will provide important insights for the research community to design PVSK materials to achieve record PCEs accompanied by high stability and longevity.

## Methods

**Synthesis of phenylethylammonium iodide**. In a typical synthesis, 4.8 g of phenethylamine (39.6 mmol, Aldrich, >99%) was dissolved in 15 mL of ethanol and placed in iced bath. Under vigorous stirring, 10.8 g of hydroiodic acid (57 wt% in H$_2$O, 48.1 mmol, Sigma-Aldrich, 99.99%, contains no stabilizer) was slowly added to the solution. The solution was stirred overnight to ensure complete reaction, which was followed by removal of the solvent by a rotary evaporator. The resulting solid was washed with diethyl ether several times until the color is changed to white. The white solid was further purified by recrystallization in mixed solvent of methanol and diethyl ether. Finally, white plate-like solid was filtered and dried under vacuum (yield around 90%).

**Device fabrication**. Indium doped tin oxide (ITO) glass was cleaned with successive sonication in detergent, deionized (DI) water, acetone and 2-propanol for 15 min, respectively. The cleaned substrates were further treated with UV-ozone to remove the organic residual and enhance the wettability. A total of 30 mM SnCl$_2$·2H$_2$O (Aldrich, >99.995%) solution was prepared in ethanol (anhydrous, Decon Laboratories Inc.), which was filtered by 0.2 μm syringe filter before use. To form a SnO$_2$ layer, the solution was spin-coated on the cleaned substrate at 3000 rpm for 30 s, which was heat-treated at 150 °C for 30 min. After cooling down to room temperature, the spin-coating process was repeated one more time, which was followed by annealing at 150 °C for 5 min and 180 °C for 1 h. The SnO$_2$ coated ITO glass was further treated with UV-ozone before spin-coating of PVSK solution. The PVSK layer was prepared by the modified adduct method[26]. The bare FAPbI$_3$ layer was formed from the PVSK solution containing equimolar amount of HC(NH$_2$)$_2$I (FAI, Dyesol), PbI$_2$ (TCI, 99.99%) and N-Methyl-2-pyrrolidone (NMP, Sigma-Aldrich, anhydrous, 99.5%) in N,N-Dimethylformamide (DMF, Sigma-Aldrich, anhydrous, 99.8%). Typically, 172 mg of FAI, 461 mg of PbI$_2$ and 99 mg of NMP were added to 600 mg of DMF. For the 2D PVSK (PEA$_2$PbI$_4$) and Cs incorporated PVSK, corresponding amount of FAI was replaced with PEAI and CsI. For example, FAPbI$_3$ with 1.67 mol% PEA$_2$PbI$_4$ PVSK was formed from the precursor solution containing 166.4 mg of FAI, 8.2 mg of phenylethylammonium

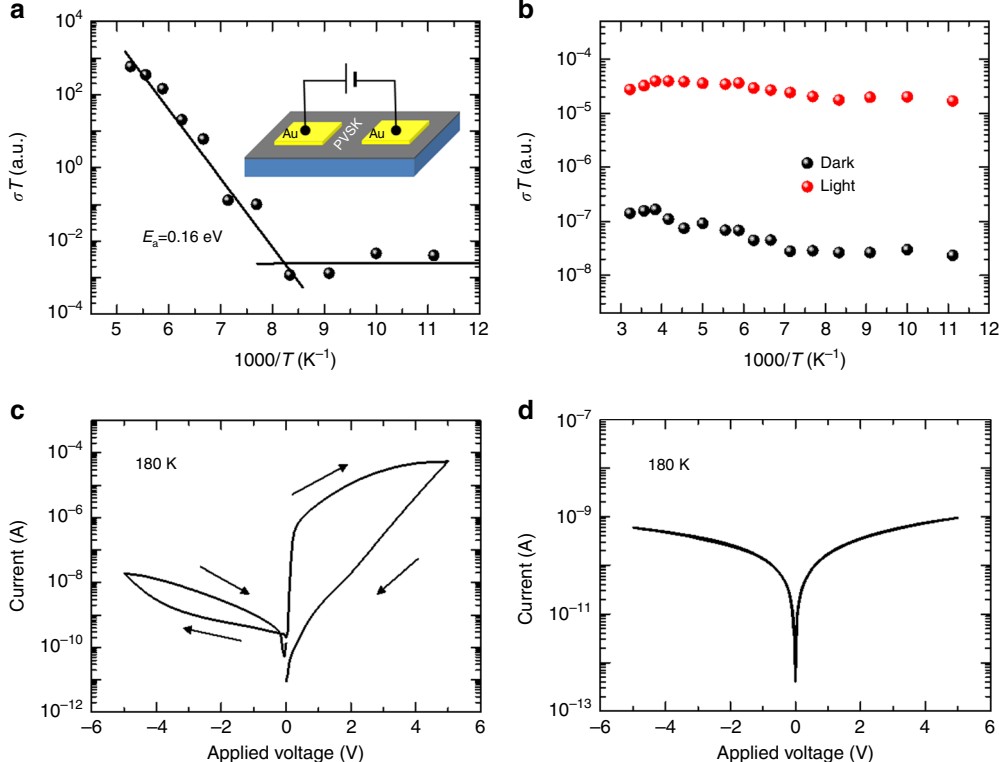

**Fig. 6** Suppressed ion migration with 2D perovskite. Temperature-dependent conductivity of **a** bare FAPbI$_3$ film and **b** with 1.67 mol% 2D PEA$_2$PbI$_4$ perovskite. Red circles in **b** indicate the data measured under moderate light illumination (intensity lower than 10 mW cm$^{-2}$). Current–voltage curves measured from the devices at 180 K. **c** Bare FAPbI$_3$ film and **d** with 1.67 mol% 2D PEA$_2$PbI$_4$ perovskite

iodide (PEAI), 453.4 mg of PbI$_2$ and 97.4 mg of NMP in 600 mg of DMF. With 2 mol% of Cs, the precursor solution was prepared by mixing 163.0 mg of FAI, 8.2 mg of PEAI, 5.0 mg of CsI (Alfa Aesar, 99.999%), 453.4 mg of PbI$_2$ and 97.4 mg of NMP in 600 mg. For the best performing devices in Fig. 2d, the amount of DMF was adjusted to 550 mg. Spin-coating of PVSK and hole transporting layer was performed in a glove box filled with dry air. The PVSK solution was spin-coated at 4000 rpm for 20 s where 0.15 mL of diethyl ether (anhydrous, >99.0%, contains BHT as stabilizer, Sigma-Aldrich) was dropped after 10 s on the spinning substrate. The resulting transparent adduct film was heat-treated at 100 °C for 1 min followed by 150 °C for 10 min. (for the best performing target device, the annealing condition was adjusted to 80 °C 1 min followed by 150 °C for 20 min) The spiro-MeOTAD solution was prepared by dissolving 85.8 mg of spiro-MeOTAD (Lumtec) in 1 mL of chlorobenzene (anhydrous, 99.8%, Sigma-Aldrich) which was doped by 33.8 μl of 4-tert-butylpyridine (96%, Aldrich) and 19.3 μl of Li-TFSI (99.95%, Aldrich, 520 mg mL$^{-1}$ in acetonitrile) solution. The spiro-MeOTAD solution was spin-coated on the PVSK layer at 3000 rpm for 20 s by dropping 17 μl of the solution on the spinning substrate. On top of the spiro-MeOTAD layer, ca. 100 nm-thick silver or gold layer was thermally evaporated at 0.5 Ås$^{-1}$ to be used as an electrode.

**Material characterization**. The PVSK layer was coated on a SnO$_2$ coated ITO substrate for the measurements. UV-vis absorption spectra were recorded by U-4100 spectrophotometer (Hitachi) equipped with integrating sphere. The monochromatic light was incident to the substrate side. X-ray diffraction (XRD) patterns were obtained by X-ray diffractometer (PANalytical) with Cu kα radiation at a scan rate of 4° min$^{-1}$. Surface and cross-sectional morphology of the films and devices were investigated by scanning electron microscopy (SEM, Nova Nano 230). For the cross-sectional image, cross-sectional surface of the sample was coated with ca. 1 nm-thick gold using sputter to enhance the conductivity. Transmission electron microscopic (TEM) analysis was performed by Titan Krios (FEI). The PVSK film was scratched off from the substrate and dispersed in toluene by sonication for 10 min, which was dropped on an aluminium grid. Accelerating voltage of 300 kV was used for the measurement. Steady-state photoluminescence (PL) signal was analyzed by a Horiba Jobin Yvon system. A 640 nm monochromatic laser was used as an excitation fluorescence source. Time resolved PL decay profiles were obtained using a Picoharp 300 with time-correlated single-photon counting capabilities. The films were excited by a 640 nm pulse laser with a repetition frequency of 100 kHz provided by a picosecond laser diode head (PLD 800B, PicoQuant). The energy density of the excitation light was ca. 1.4 nJ cm$^{-2}$, in which carrier annihilation and non-geminate recombination are negligible[34]. Ultraviolet photoelectron

spectroscopic (UPS) analysis was carried out using Kratos Ultraviolet photoelectron spectrometer. He I (21.22 eV) source was used as an excitation source. The PVSK films were coated on ITO substrate and grounded using silver paste to avoid the charging during the measurement. Conductive atomic force microscopic (AFM) measurement was performed by Bruker Dimension Icon Scanning Probe Microscope equipped with TUNA application module. The TUNA module provides ultra-high tunneling current sensitivity (<1 pA) with high-lateral resolution. Antomony doped Si tip (0.01–0.025 Ohm-cm) coated with 20 nm Pt-Ir was used as a probe. To avoid the electrically driven degradation during the measurement, low bias voltage (100 mV) was applied. The measurement was carried out under either room right or low intensity light illumination provided by AFM setup. The temperature-dependent conductivity measurement was carried out using a commercial probe station (Lakeshore, TTP4) in which temperature of the device was controlled by thermoelectric plate and flow of liquid nitrogen. The electrical measurement was conducted with a source/measurement unit (Agilent, B2902A).

**Device characterization**. Current density–voltage ($J$–$V$) curves of the devices were measured using Keithley 2401 source meter under simulated one sun illumination (AM 1.5G, 100 mW cm$^{-2}$) in ambient atmosphere. The one sun illumination was generated from Oriel Sol3A with class AAA solar simulator (Newport), in which light intensity was calibrated by NREL-certified Si photodiode equipped with KG-5 filter. Typically, the $J$–$V$ curves were recorded at 0.1 Vs$^{-1}$ (between 1.2 V and −0.1 V with 65 data points and 0.2 s of delay time per point). During the measurement, the device was covered with metal aperture (0.100 cm$^2$) to define the active area. All the devices were measured without pre-conditioning such as light-soaking and applied bias voltage. Steady-state power conversion efficiency was calculated by measuring stabilized photocurrent density under constant bias voltage. The external quantum efficiency (EQE) was measured using specially designed system (Enli tech) under AC mode (frequency = 133 Hz) without bias light. For electroluminescence measurement, a Keithley 2400 source meter and silicon photodiode (Hamamatsu S1133-14, Japan) were used to measure Current–voltage–luminance characteristics of PVSK solar cells. Electroluminescence spectra were recorded by Horiba Jobin Yvon system, and used to calculate radiance and external quantum efficiency of PVSK solar cells. All the devices were assumed as Lambertian emitter in the calculation.

**Stability test**. Moisture stability of the films was tested by exposing the PVSK films under relative humidity of 80 ± 5% and room light. Absorbance of the films was measured every 2 h while XRD of the films were recorded every 12 h. For the devices, ex-situ test was conducted by storing the devices in desiccator (relative

humidity, RH <30%) under dark condition. The device was taken out and measured in ambient condition. For operational stability, maximum power point (MPP) tracking and continuous light exposure under open-circuit condition were performed in ambient condition (RH around 50%, $T$ around 40 °C). For the MPP tracking, the photocurrent density was monitored while the devices were biased at MPP under 1 sun illumination. For light exposure under open-circuit condition, the encapsulated devices were exposed to ca. 0.9 sun (90 ± 5 mW cm$^{-2}$) generated by halogen lamps under open-circuit condition, of which steady-state PCE was periodically measured with different exposure time under 1 sun illumination. The encapsulation of the device was performed inside the glove box filled with nitrogen by using an UV-curable adhesive and a piece of glass. The glass substrate was superimposed on active layer and fixed with the UV-curable adhesive.

**Data availability**. The authors declare that the data supporting the findings of this study are available within the paper and its supplementary information files.

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

## Acknowledgements

This work was supported by the Air Force Office of Scientific Research (AFOSR, Grant No. FA9550-15-1-0333), Office of Naval Research (ONR, Grant No. N00014-17-1-2484), National Science Foundation (NSF, Grant No. ECCS-EPMD-1509955), and Horizon PV.

## Author contributions

J.-W.L. conceived an idea and led overall project under supervision of Y.Y. J.-W.L. and Z.D. fabricated devices and characterized the materials. T.-H.H. assisted the device fabrication and performed EL measurement. C.C. and Y.H. carried out TEM characterization. S.-Y.C. helped UPS measurement. S.L. performed temperature-dependent conductivity measurement. H.Z. synthesized the PEAI. N.D.M. and P.S.

commented on the manuscript. All the authors discussed and commented on the manuscript.

## Additional information

**Competing interests:** The authors declare no competing interests.

