## [Peer Review File · Nature Communications]

Reviewers' Comments:

Reviewer #1:

Remarks to the Author:

Authors report a study on stabilizing FAPbI₃ phase by adding 2D PEA perovskite. They managed to get certified 19.77% stabilized efficiency with a low Voc loss of 0.35V. However, the novelty of this work is questionable. FAPbI₃ is known to be stabilized by PEA or other 2D perovskites (DOI: 10.1002/aenm.201601307; DOI: 10.1021/acs.nanolett.7b01500) and can be further stabilized by Cs incorporation. Thus, stability can be expected to be improved by adding PEA⁺ and Cs⁺ in FAPbI₃. In this regards, it is hard to find new science in this work. In addition, authors only show several hours of device stability under operational conditions, which is far from long-term stability test. Nevertheless, this work may be reconsidered if authors describe distinctly the important point differentiating the reported results and provide long term stability.

1. How does this system different from the previous PEA stabilized FAPbI₃ perovskite paper (DOI: 10.1002/aenm.201601307) reporting 17.7% efficiency by Jen group? Snaith group has added lower dimensional 2D perovskites into a Cs-FA perovskite system to improve Voc and stability (doi:10.1038/nenergy.2017.135). They also revealed that the 2D perovskites preferentially align at grain boundaries. Similar effects have also been observed by many other groups (DOI: 10.1002/aenm.201701048; DOI: 10.1002/aenm.201701038). Can the authors compare their work with those efforts?
2. Adding Cs⁺ into FAPbI₃ to stabilize the black phase has already been revealed by Snaith (DOI: 10.1126/science.aad5845) and Graztel groups (DOI: 10.1039/C5EE03874J). It is important that the authors present a strong science argument with well supported data.
3. Authors address "The blue-shift of the PL peak might be due to formation of a quasi-3D PVSK, where the 2D PVSK possibly incorporated into the lattice of the 3D PVSK to induce carrier confinement effects." It is impossible for 2D PVSK to incorporate into the lattice of the 3D PVSK. They have completely different lattice parameters and cannot merge into a single crystal unit.
4. It is hard to draw a solid conclusion on the 2D perovskite by merely judging the PL spectra as charge transfer may occur between the 2D and 3D perovskite. Authors need to provide absorption spectra as well as other data to support their conclusions.
5. The moisture stability is tested under only 30% relative humidity and the operational stability is done for only 450 min. More rigorous stability test, e.g. 85% relative humidity and/or 1000 hour operational stability would improve the quality of the paper.

Reviewer #2:

Remarks to the Author:

The balanced stability and high performance has been now the main challenge for the development and commercialization of perovskite. The FAPbI₃ with intrinsic higher thermal stability are promising but its performance and moisture stability are challenging. The authors here present a method for preparing 2D-3D perovskite solar cells combined with Cs incorporation for the improvement of FAPbI₃ films on stability and enhanced PV performance. In general, this paper is interesting and well-written and I would recommend it for publication at Nature Communications once the author address the following comments.

1. The photos for 8 h 12 h 16 h 20 h 24 h in Figure 3a is total unclear. Figure caption for Fig 1c should add discription for PL result.
2. Some previous reported 2D-3D perovskite to stablize the FAPbI₃ and CsPbI₃ (Nature Energy, 2017, 2, 17135; Sci. Adv. 2017, 3, e1700841.) should be cited.
3. Both previous report of Nature Energy, 2017, 2, 17135 and this work show that the 2D perovskite could be vertically located at grain boundary of FAPbI₃ as Ruddlesden-Popper perovskite while it usually form a homologous 2D-3D perovskite in MAPbI₃ and its is so difficult to obtain such 2D Ruddlesden-Popper for MAPbI₃. Could the authors give more thorough discussion on the reason why there is such difference between FACs and MA cation based perovskite.

4. The label difference between w/2D PVSK and w/2D PVSK and Cs should be modified, currently they are too similar to each other.
5. In the stability test, the author mentioned about the RH but it is better to include the temperature too because the temperature can significant affect the absolute humidity value.

Response to Reviewer Comments

Reviewers' comments:

Reviewer #1 (Remarks to the Author):

Authors report a study on stabilizing FAPbI₃ phase by adding 2D PEA perovskite. They managed to get certified 19.77% stabilized efficiency with a low Voc loss of 0.35V. However, the novelty of this work is questionable. FAPbI₃ is known to be stabilized by PEA or other 2D perovskites (DOI: 10.1002/aenm.201601307; DOI: 10.1021/acs.nanolett.7b01500) and can be further stabilized by Cs incorporation. Thus, stability can be expected to be improved by adding PEA⁺ and Cs⁺ in FAPbI₃. In this regards, it is hard to find new science in this work. In addition, authors only show several hours of device stability under operational conditions, which is far from long-term stability test. Nevertheless, this work may be reconsidered if authors describe distinctly the important point differentiating the reported results and provide long term stability.

Ans) We would like to thank the reviewers for the valuable comments. The comments were greatly helpful to improve the quality of our manuscript.

1. How does this system different from the previous PEA stabilized FAPbI₃ perovskite paper (DOI: 10.1002/aenm.201601307) reporting 17.7% efficiency by Jen group? Snaith group has added lower dimensional 2D perovskites into a Cs-FA perovskite system to improve Voc and stability (doi:10.1038/nenergy.2017.135). They also revealed that the 2D perovskites preferentially align at grain boundaries. Similar effects have also been observed by many other groups (DOI: 10.1002/aenm.201701048; DOI: 10.1002/aenm.201701038). Can the authors compare their work with those efforts?

Ans) We thank the reviewer for the constructive criticism. We have added the papers mentioned by reviewer as references in the revised manuscript. We were also aware of the important previous works introducing 3D-2D mixed perovskite systems incorporating formamidinium perovskites. However, we believe there has been no effective way to realize the competitive low bandgap FAPbI₃ (or FACsPbI₃) perovskite solar cells that can compete with those based on bare MAPbI₃ or mixed cation-halide perovskites in terms of both stability and performance. In this aspect, we believe our report provides a very effective way to achieve efficient and stable perovskite solar cells based on 3D-2D perovskites (with

identical bandgap to FAPbI₃, please also see the answer for question 2) that can readily compete with 3D perovskite solar cells. Also, thanks to the reviewer's constructive comments, we believe the revised manuscript provides important clues for elucidating the origin of the improved performance and stability in 3D-2D perovskite solar cells.

2. Adding Cs⁺ into FAPbI₃ to stabilize the black phase has already been revealed by Snaith (DOI: 10.1126/science.aad5845) and Graztel groups (DOI: 10.1039/C5EE03874J). It is important that the authors present a strong science argument with well supported data.

Ans) We appreciate the comments by the reviewer. As the reviewer pointed out, the stabilization of the black phase of formamidinium perovskite has been previously reported by several groups (N.-G. Park et al. *Adv. Energy Mater.* 2015, 5, 1501310. H. J. Snaith et al, *Science*, 2016, 351, 151. M. Graztel et al. *Energy Environ. Sci.* 2016, 9, 1989.) Typically, at least 10 mol% replacement of FA with Cs (5 mol% in combination with 17 mol% of MA) is required to stabilize the black perovskite phase in case of pure FAPbI₃. As mentioned in our original manuscript and supplementary information, we tried to minimize the incorporation of smaller ions such as Cs⁺, MA⁺, Br⁻ to maintain the low bandgap of pure FAPbI₃ perovskite. Indeed, we showed that replacement of 2 mol% of FA⁺ with Cs⁺ does not have significant impacts on the absorption onset of the perovskite films (**Supplementary Figure 9** and **note 2**). Also, the film without 2 mol% Cs but with 1.67 mol% of 2D perovskite was also stable under relative humidity of 80±5% for 24 h as seen in **Figure 3a** and **b**. We believe that the 2 mol% replacement of FA with Cs reduces the defect density as observed from time-resolved photoluminescence in **Figure 2**, but it does not play another critical role in our system.

For example, we compared phase-stability of the FAPbI₃ and FA_{0.98}Cs_{0.02}PbI₃ perovskite film under a relative humidity of 70±5% (**Figure R1**). As can be seen in the photos and X-ray diffraction (XRD) patterns of the films, both of FAPbI₃ and FA_{0.98}Cs_{0.02}PbI₃ perovskite films rapidly degraded within 24 h. The black color of the films were rapidly changed to yellow, which can be attributed to conversion of the cubic perovskite phase to yellow hexagonal non-perovskite phase. Therefore, we concluded that the significantly improved phase-stability is mainly owed to the added 2D perovskite.

Figure R1| Humidity stability of FAPbI₃ and FA_{0.98}Cs_{0.02}PbI₃ perovskite films. a, photos and b, c, X-ray diffraction patterns of the FAPbI₃ and FA_{0.98}Cs_{0.02}PbI₃ perovskite films with exposure to relative humidity (RH) of 70±5% (T=20±2 °) for different time. Peaks from cubic FAPbI₃ phase were indexed by α whereas δ and * indicate hexagonal FAPbI₃ and PbI₂, respectively.

We also compared the steady-state and time-resolved photoluminescence (PL) properties of the FAPbI₃ and FA_{0.98}Cs_{0.02}PbI₃ perovskite films in **Figure R2**. As seen in **Figure R2a**, the peak PL intensity of bare FA_{0.98}Cs_{0.02}PbI₃ is 9.39×10^5 which was improved to 1.17×10^6 with 2 mol% of Cs. The PL lifetime of FAPbI₃ (40.4 ns) was prolonged to 52.0 ns with incorporation of 2 mol% Cs. However, the improvement in PL intensity and lifetime is much less compared to that observed with addition of 2D perovskite (~5 times and ~10 times enhancement in PL intensity and lifetime, respectively). Therefore, we concluded that improvement of the phase-stability and optoelectronic quality of the perovskite film is predominantly due to the addition of 2D perovskite.

Figure R2| Photoluminescence (PL) properties of FAPbI₃ and FA_{0.98}Cs_{0.02}PbI₃ perovskite films. a, Steady-state and **b,** time-resolved PL measurements of the FAPbI₃ and FA_{0.98}Cs_{0.02}PbI₃ perovskite films.

The **Figure R1** and **R2** were added to the supplementary information as **Figure S11** and **Figure S12**. Also, a corresponding description was added to the supplementary information as **supplementary note 3**.

We believe the added 2D perovskite not only protects the 3D perovskite from moisture but also reduces the formation energy of the cubic perovskite phase in consideration of the enhanced phase-purity of the resulting film (**Figure 1b**). We performed a closer inspection on the normalized X-ray diffraction (XRD) patterns of the perovskite films with different amounts of added 2D perovskite (**Figure R3**) to find any correlations between the added 2D perovskite and crystal structure of the FAPbI₃. Interestingly, a systematic change in peak position was observed with different amounts of added 2D perovskite for which the XRD peaks were slightly shifted towards higher angles with the addition of relatively smaller amounts of 2D perovskite (1.25, 1.67, 2.50 and 5.0 mol%), indicating that the lattice constant of FAPbI₃ is reduced, probably as a result of compressive strain associated with the 2D perovskite. We speculate that the reduction in the lattice constant can be related to the enhanced phase-purity of cubic FAPbI₃ as it will have equivalent effects with incorporation of smaller ‘A’ site cations on tolerance factor and thus enthalpy of formation.¹ With addition of 10 mol% 2D perovskite, the peak rather shifted to lower angles with an additional peak appearing (**Supplementary Figure 4**), which is probably associated with the formation of quasi-3D perovskite (please see also answer of question 4).

Figure R3 | Normalized X-ray diffraction patterns of FAPbI₃ perovskite films with different amount of added 2D PEA₂PbI₄ perovskite. **a**, full spectra and **b**, magnified (001) orientation peaks

Figure R3 was added to the supplementary information as **Figure S6** and the following sentences were added to the revised manuscript in page 4. “A closer inspection on the normalized X-ray diffraction (XRD) patterns of the PVSK films with different amounts of added 2D PVSK (**Supplementary Fig. 6**) was taken to find any correlations between the added 2D PVSK and crystal structure of FAPbI₃. Interestingly, a systematic change in peak position was observed with different amounts of 2D PVSK for which the XRD peaks were slightly shifted towards higher angles with the addition of relatively smaller amounts of 2D PVSK (1.25, 1.67, 2.50 and 5.0 mol%). This indicates that the lattice constant of FAPbI₃ is reduced, likely due to compressive strain associated with the added 2D PVSK. We speculate that the reduction in lattice constant can also be related to the enhanced phase-purity of cubic FAPbI₃ as it will have equivalent effects with incorporation of smaller ‘A’ site cations on the tolerance factor and thus enthalpy of formation.¹”

3. Authors address “The blue-shift of the PL peak might be due to formation of a quasi-3D PVSK, where the 2D PVSK possibly incorporated into the lattice of the 3D PVSK to induce carrier confinement effects.” It is impossible for 2D PVSK to incorporate into the lattice of the 3D PVSK. They have completely different lattice parameters and cannot merge into a single crystal unit.

Ans) We think our ambiguous writing caused the misunderstanding. As the reviewer pointed out, the 2D perovskite is not incorporated into the lattice of 3D perovskite but is mixed with

3D perovskite to form quasi-3D perovskite. To clarify this point, we modified following sentence “The blue-shift of the PL peak might be due to formation of a quasi-3D PVSK, where the 2D PVSK possibly incorporated into the lattice of the 3D PVSK to induce carrier confinement effects.” to “The blue-shift of the PL peak might be due to formation of a quasi-3D PVSK, where charge carriers are confined by large potential barrier originated from the 2D PVSK.”

4. It is hard to draw a solid conclusion on the 2D perovskite by merely judging the PL spectra as charge transfer may occur between the 2D and 3D perovskite. Authors need to provide absorption spectra as well as other data to support their conclusions.

Ans) We thank the reviewer for the comment. As the reviewer pointed out, there will be a charge transfer between the 2D and 3D perovskites. Identical steady-state photoluminescence (PL) spectra with addition of 2D perovskite less than 10 mol% indicates the presence of pure 3D perovskite phase in terms of its density of states (DOS, **Figure R4**), so we speculate that the 2D and 3D perovskite do not form a mixed phase (quasi-3D perovskite) in consideration of XRD spectra showing pure cubic FAPbI_3 phase. When the amount of added 2D perovskite is 10 mol%, the steady-state PL spectrum shifted towards a shorter wavelength, indicating that the DOS of the lowest energy state (3D perovskite) is affected by the added 2D perovskite. We believe this evidences the formation of quasi-3D perovskite in which 2D perovskite is intercalated into the bulk of 3D perovskite to induce the charge carrier confinement and thus increased bandgap (**Figure R4b**). This speculation is well-correlated with additional XRD peak appearing at the low angle region (**Supplementary Fig. 4f**), which typically originates from the formation of quasi-3D perovskites.² Also, significantly reduced PL lifetimes in **Supplementary Fig. 13** indicate the possible charge carrier confinement effect by the 2D perovskite.

Figure R4 | Schematic band diagram for FAPbI₃ perovskite films with different amount of added 2D PEA₂PbI₄ perovskite. **a**, smaller amount (<10 mol%) and **b**, higher amount (>10 mol%)

As the reviewer recommended, we measured the absorption spectra of the perovskite films with different amounts of added 2D perovskite to verify the change in absorption onset (Figure R5). As seen in the Figure R5a and b, all the perovskite films with 2D perovskite showed enhanced absorption in the wavelength region from 550 nm to 820 nm, which was ascribed to enhanced phase-purity. With closer inspection, a slight blue shift of the absorption onset with a decrease in absorption over whole wavelength region was identified with addition of 10 mol% of 2D perovskite, which is correlated with the blue shift of steady-state PL spectrum that can be associated with the formation of quasi-3D perovskite.

Figure R5 | Absorption spectra of FAPbI₃ perovskite films with different amount of added 2D PEA₂PbI₄ perovskite. **a**, full spectra and **b**, magnified onset region

The **Figure R5** was added to the supplementary information as **Figure S7**, and following sentences were added to the revised manuscript in page 5. “The absorption spectra with different amounts of 2D PVSK are demonstrated in **Supplementary Fig. 7**. While all the perovskite films with 2D PVSK showed enhanced absorption compared to bare FAPbI₃ films, a slight blue shift of the absorption onset with decreases in absorption over the whole wavelength region was identified with the addition of 10 mol% of 2D PVSK, which is correlated with the blue shift of the steady-state PL spectrum that can be associated with the formation of quasi-3D PVSK.”

5. The moisture stability is tested under only 30% relative humidity and the operational stability is done for only 450 min. More rigorous stability test, e.g. 85% relative humidity and/or 1000 hour operational stability would improve the quality of the paper.

Ans) We thank the reviewer for the valuable comments. Actually, the moisture test of the perovskite films without and with 2D perovskite was performed in relative humidity (RH) of 80±5% as we specified in the original manuscript. We believe this is rigorous enough to test the moisture stability of the perovskite films. For the operational stability test, we agree with reviewer`s comment that a longer period of the test is essential for evaluation of the stability.

To address the reviewer`s comment, we performed 500 h of light exposure test with the encapsulated control (bare FAPbI₃ device) and target devices (w/ 1.67 mol% 2D PVSK). The encapsulated devices were exposed to ca. 0.9 sun (90±10 mW/cm²) under open-circuit condition, of which steady-state PCE was periodically measured with different exposure time. As seen in **Figure R6c**, both of the devices showed rapid initial decay in PCEs followed by slower decay with almost linear profile, which is in agreement with previous reports.³ After 500 h of exposure, the control device was degraded to ca. 52.3% of initial PCE whereas target device maintained 72.3% of the initial PCE, indicating enhanced stability with addition of 2D perovskite. Although we could not performed longer exposure test due to limitation in our lamp lifespan, we could extract tentative T80 (time at which PCE of the device decays to 80% of initial PCE) for the devices by fitting of post-burn-in region in which PCE of the device shows almost linear decay profile (after 48 h). The T80 for control and target devices were calculated to be 592 h and 1362 h, respectively. The T80 of the devices were enhanced more than two times by incorporating the 2D perovskite.

Figure R6| Improved stability with 2D perovskite. **a**, Evolution of power conversion efficiency (PCE) of control and target devices. The devices were stored under dark with controlled humidity ($\text{RH} < 30\%$). **b**, Maximum power point tracking of the devices under 1 sun illumination in ambient condition without encapsulation. **c**, Evolution of the PCEs measured from the encapsulated control and target devices exposed to continuous light ($90 \pm 5\text{ mW/cm}^2$) under open-circuit condition. The stabilized PCEs were measured at each time. Initial stabilized PCEs for control and target devices were 14.5% and 17.5%, respectively. The broken lines are linear fit of the post burn-in region (after 48 h)

Figure 5 in original manuscript was replaced with **Figure R6**, and following sentences were added to the revised manuscript in page 11. “We performed 500 h of light exposure test with the encapsulated control (bare FAPbI_3 device) and target devices (w/ 1.67 mol% 2D PVSK). The encapsulated devices were exposed to ca. 0.9 sun ($90 \pm 10\text{ mW/cm}^2$) under open-circuit condition, of which the steady-state PCE was periodically measured for different exposure time. As seen in **Figure 5c**, both of the devices showed a rapid initial decay in PCE followed by slower decay with an almost linear profile, which is in agreement with previous reports.³ After 500 h of exposure, the control device degraded to ca. 52.3% of its initial PCE whereas the target device maintained 72.3% of the initial PCE, indicating enhanced stability with addition of 2D PVSK. We could extract tentative T80 (time at which PCE of the device

decays to 80% of initial PCE) for the devices by fitting of the post-burn-in region in which the PCE of the device shows an almost linear decay profile (after 48 h). The T80 for control and target devices were calculated to be 592 h and 1362 h, respectively.”

In addition to this, we also performed maximum power point (MPP) tracking of the target device for 130 h since the device will be biased at maximum power point under real operating conditions. 18.7% of the initial PCE was degraded for 130 h of operation, which is relatively slower compared to the device maintained at open-circuit condition. This is correlated with previous studies that attributed the faster degradation under open-circuit condition to larger number of photo-generated charge carriers recombining within the device.⁴

Figure R7| Maximum power point tracking of the target device under 1 sun illumination in ambient condition with encapsulation. Inset of the figure shows the PCE without normalization.

Figure R7 was added to the supplementary information as **Figure S28**, and the following sentences were added to the revised manuscript in page X. “We also performed MPP tracking of the encapsulated target device under 1 sun (100 mW/cm^2) illumination in **Supplementary Fig. 28**. 18.7% of initial PCE was degraded for 130 h of operation, which is relatively slower compared to the device maintained at open-circuit condition. This is correlated with previous studies that attributed the faster degradation under open-circuit condition to larger number of photo-generated charge carriers recombining within the device.⁴”

Under operational condition with abundant photo-generated charges and built-in electric field, the major factors causing the degradation of the devices might be the highly mobile and reactive charged defects (ions) and/or trapped charge carriers associated with it.^{3,5} We suppose that migration of the charged defects or ions is possibly suppressed by 2D perovskite at grain boundaries. We performed temperature-dependent conductivity (σ) measurement of the lateral devices (inset of **Figure R8** describes the structure of the devices) to evaluate the activation energy for ion migration (**Figure R8**). The activation energy (E_a) for the migration can be determined according to the Nernst-Einstein relation:³

$$\sigma(T) = \frac{\sigma_0}{T} \exp\left(\frac{-E_a}{kT}\right)$$

where k is Boltzmann constant, σ_0 is a constant. With bare FAPbI₃ perovskite, exponential enhancement in conductivity was clearly identified at around 130 K (**Figure R8a**), which is attributed to contribution of ions. The E_a for bare FAPbI₃ film was calculated to be 0.16 eV, indicating significant contribution of activated ions at room temperature, which might cause degradation of the material and device under operational condition with built-in electric field. The pronounced current-voltage hysteresis behavior was observed even at very low temperature (180 K, **Figure R8c**). For addition of 2D perovskite, the film did not show noticeable enhancement in conductivity with increased temperature although the overall conductivity was relatively lower than the bare FAPbI₃ film (**Figure R8c**). Even with moderate light illumination, it does not show indication of activated ions. As a result, the current-voltage curve did not show any hysteresis behavior (**Figure R8d**). We believe the suppressed ion migration contributes to enhanced operational stability of the target device.

Figure R8 | Temperature-dependent conductivity of **a**, bare FAPbI₃ film and **c**, with 1.67 mol% 2D PEA₂PbI₄ perovskite. Current-voltage curves measured from the devices at 180 K. **c**, bare FAPbI₃ film and **d**, with 1.67 mol% 2D PEA₂PbI₄ perovskite.

The **Figure R8** was added to supplementary information as **Figure S29**, and the following sentences were added to the revised manuscript. “Under operational condition with abundant photo-generated charges and built-in electric field, the major factors causing the degradation of the devices might be the highly mobile and reactive charged defects (ions) and/or trapped charge carrier associated with it.^{3,5} We suppose that migration of the charged defects or ions is possibly suppressed by 2D PVSK at grain boundaries. The temperature-dependent conductivity (σ) measurement of the lateral devices was performed to evaluate the activation energy for the ion migration (**Supplementary Fig. 29**). The activation energy (E_a) for the migration can be determined according to the Nernst-Einstein relation,³

$$\sigma(T) = \frac{\sigma_0}{T} \exp\left(\frac{-E_a}{kT}\right)$$

where k is Boltzmann constant, σ_0 is a constant. Inset of **Supplementary Fig. 29a** describes the structure of the lateral devices. With bare FAPbI₃ PVSK, exponential enhancement in conductivity was clearly identified at around 130 K (**Supplementary Fig. 29a**), which is attributed to contribution of ions. The E_a for bare FAPbI₃ film was calculated to be 0.16 eV, indicating significant contribution of activated ions at room temperature, which might cause degradation of the material and device under operational condition with built-in voltage. The pronounced current-voltage hysteresis behavior was observed even at very low temperature (180 K, **Supplementary Fig. 29c**). In case of the PVSK film with 2D PVSK, the film did not show noticeable enhancement in conductivity with increased temperature although the overall conductivity was relatively lower than the bare FAPbI₃ film (**Supplementary Fig. 29c**). Even with moderate light illumination, it does not show the indicative of activated ions. As a result, the current-voltage curve did not show any hysteresis behavior (**Supplementary Fig. 29d**). We believe the suppressed ion migration contributes to enhanced operational stability of the target device.”

Reviewer #2 (Remarks to the Author):

The balanced stability and high performance has been now the main challenge for the development and commercialization of perovskite. The FAPbI₃ with intrinsic higher thermal stability are promising but its performance and moisture stability are challenging. The authors here present a method for preparing 2D-3D perovskite solar cells combined with Cs incorporation for the improvement of FAPbI₃ films on stability and enhanced PV performance. In general, this paper is interesting and well-written and I would recommend it for publication at Nature Communications once the author address the following comments.

Ans) We would like to thank the reviewers for the valuable comments. The comments were greatly helpful to improve the quality of our manuscript.

1. The photos for 8 h 12 h 16 h 20 h 24 h in Figure 3a is total unclear. Figure caption for Figure 1c should add description for PL result.

Ans) We thank the reviewer for the detailed comment. **Figure 3a** was corrected and the caption of **Figure 1c** was added.

2. Some previous reported 2D-3D perovskite to stabilize the FAPbI₃ and CsPbI₃ (Nature Energy, 2017, 2, 17135; Sci. Adv. 2017, 3, e1700841.) should be cited.

Ans) We have added the corresponding papers as references in the revised manuscript.

3. Both previous report of Nature Energy, 2017, 2, 17135 and this work show that the 2D perovskite could be vertically located at grain boundary of FAPbI₃ as Ruddlesden-Popper perovskite while it usually form a homologous 2D-3D perovskite in MAPbI₃ and it is so difficult to obtain such 2D Ruddlesden-Popper for MAPbI₃. Could the authors give more thorough discussion on the reason why there is such difference between FACs and MA cation based perovskite.

Ans) We thank the reviewer for pointing out the important aspects of 2D-3D perovskites. We also tried to incorporate the 2D PEA₂PbI₄ perovskite into the MAPbI₃ perovskite. We observed slight enhancement in peak intensity of X-ray diffraction pattern, and do not see any additional peaks from the XRD pattern as observed in FAPbI₃ perovskite (**Figure R9**). Also, the normalized PL spectra of the films were almost identical regardless of the added 2D PVSK (inset of **Figure R10a**). Therefore, we speculate that the added 2D PVSK is probably formed at the grain boundaries. However, we observed very different phenomena with the case of FAPbI₃ (or FACsPbI₃) perovskite. First, the diffraction peaks of MAPbI₃ films were shifted toward lower angle upon addition of 1.67 mol% 2D PEA₂PbI₄ perovskite, which is contrary to FAPbI₃ case (please see answer of question 2 by reviewer 1). Furthermore, while the steady-state PL intensity is increased with addition of 2D PVSK, the PL lifetime was decreased with 2D PVSK (61.1 ns for bare MAPbI₃ and 37.0 ns with 2D PVSK). We also fabricated the devices with bare MAPbI₃ and with 1.67 mol% 2D perovskite. However, all the photovoltaic parameters except open-circuit voltage (V_{OC}) were degraded with addition of 2D PVSK. We believe these are indicative of the different interactions between the MAPbI₃ and 2D PEA₂PbI₄ perovskite with those of FAPbI₃ case, probably in terms of crystallography, causing completely different effects on electronic properties. We could not find the possible explanation yet, but plan to perform more in-depth study on these aspects.

Figure R9 | X-ray diffraction patterns for MAPbI₃ perovskite films without and with 1.67 mol% 2D PEA₂PbI₄ perovskite. **a**, full spectra and **b**, magnified (110) orientation peaks

Figure R10 | **a**, Steady-state and **b**, time-resolved photoluminescence (PL) of MAPbI₃ perovskite films without and with 1.67 mol% 2D PEA₂PbI₄ perovskite. Inset of **a** shows the normalized PL spectra.

Figure R11 | Photovoltaic parameters measured from MAPbI₃ perovskite solar cells without and with 1.67 mol% 2D PEA₂PbI₄ perovskite **a**, Short-circuit current density (J_{sc}), **b**, Open-circuit voltage (V_{oc}), **c**, fill factor (FF) and **d**, power conversion efficiency (PCE).

4. The label difference between w/2D PVSK and w/2D PVSK and Cs should be modified, currently they are too similar to each other.

Ans) We appreciate the reviewer`s comment. We have modified the label for the w/2D PVSK.

5. In the stability test, the author mentioned about the RH but it is better to include the temperature too because the temperature can significant affect the absolute humidity value.

Ans) We thank the reviewer for the valuable comment. As the reviewer mentioned, the temperature is an important factor as we only stated relative humidity. Now we clarify the temperature for all the stability measurement.

References

- 1 Nagabhushana, G., Shivaramaiah, R. & Navrotsky, A. Direct calorimetric verification of thermodynamic instability of lead halide hybrid perovskites. *Proceedings of the National Academy of Sciences* **113**, 7717-7721 (2016).
- 2 Stoumpos, C. C. *et al.* High members of the 2D Ruddlesden-Popper halide perovskites: synthesis, optical properties, and solar cells of $(\text{CH}_3(\text{CH}_2)_3\text{NH}_3)_2(\text{CH}_3\text{NH}_3)_4\text{Pb}_5\text{I}_{16}$. *Chem* **2**, 427-440 (2017).
- 3 Domanski, K. *et al.* Migration of cations induces reversible performance losses over day/night cycling in perovskite solar cells. *Energy Environ. Sci.* **10**, 604-613 (2017).
- 4 Domanski, K., Alharbi, E. A., Hagfeldt, A., Grätzel, M. & Tress, W. Systematic investigation of the impact of operation conditions on the degradation behaviour of perovskite solar cells. *Nat. Energy* **3**, 61-67 (2018).
- 5 Ahn, N. *et al.* Trapped charge-driven degradation of perovskite solar cells. *Nat. Commun.* **7**, 13422 (2016).

Reviewers' Comments:

Reviewer #1:

Remarks to the Author:

The paper has been largely improved. However, I still have a couple of concerns:

1. Why the enhanced phase purity would cause a red-shift in the absorption spectra, but no shift in the PL? What are the impurities/less-pure phases?
2. The observation of suppressed ion migration with 2D perovskite is very important. Maybe adding this in the main text? Also, does the improved phase purity have any impact on the reduced ion migration?

Reviewer #2:

Remarks to the Author:

The authors has replied my comments and others properly. Accept it.

Response to Reviewer Comments

Reviewers' comments:

Reviewer #1 (Remarks to the Author):

The paper has been largely improved. However, I still have a couple of concerns:

1. Why the enhanced phase purity would cause a red-shift in the absorption spectra, but no shift in the PL? What are the impurities/less-pure phases?

Ans) As we mentioned in our manuscript, absorption onset of pure FAPbI₃ and FAPbI₃ with 1.67 mol% 2D perovskite is actually same, which indicates bandgap is not affected by the added 2D perovskite, so the photoluminescence spectra are almost identical. We have added inset in **Figure 1c** to show onset region of the absorption spectra with linear approximation (see **Figure R1**).

As the reviewer pointed out, absorption of the FAPbI₃ film with addition of 1.67 mol% 2D perovskite was significantly enhanced (**Figure 1c**), which was attributed to the enhanced phase-purity. The presence of a secondary phase in the pure FAPbI₃ film was identified from X-ray diffraction pattern in **Figure 1b**, in which the hexagonal non-perovskite phase was indexed by δ . We already mentioned it in the manuscript with corresponding references (page 4, reference 29). In addition, the enhancement of absorption at longer wavelength region can be partially ascribed to an enhanced light scattering with larger grain size.¹ We have added the following discussion and reference in the revised manuscript page 5.

“The enhanced absorption as seen when the 2D PVSK was added is probably due to an enhanced phase purity of the FAPbI₃, with partial contribution from an enhanced light scattering owing to the improved crystallinity.¹”

Figure R1. Absorption and normalized PL spectra of bare FAPbI₃ and FAPbI₃ with 1.67 mol% PEA₂PbI₄ 2D perovskite. Inset shows onset region of the absorption spectra with linear approximation.

2. The observation of suppressed ion migration with 2D perovskite is very important. Maybe adding this in the main text? Also, does the improved phase purity have any impact on the reduced ion migration?

Ans) We thank the reviewer's suggestion. According to the suggestion, we have added the temperature-dependent conductivity measurement in the main text as **Figure 6**. We think that passivation of grain boundaries by 2D perovskite mainly contributes to the suppressed ion migration. The grain boundaries of 3D perovskite were reported to be a major pathway for migration of ions,² passivating the grain boundaries by incorporation of the ion-migration-immune 2D PVSK likely suppressed overall ion migration in the target device.³ However, we do think that the improved phase purity of the film might also partially contribute to the suppressed ion migration because the secondary phase can generate defect sites that can act as an additional pathway for ion migration. We have added following discussion in page 13.

“As the grain boundaries of 3D perovskite were reported to be a major pathway for the migration of ions,² passivating the grain boundaries by incorporation of the ion-migration-immune 2D PVSK likely suppressed overall ion migration in the target device.³ In addition, the improved phase purity of the film might also partially contribute to the suppressed ion

migration because the secondary phase can generate defect sites that can act as an additional pathway for ion migration.”

Reviewer #2 (Remarks to the Author):

The authors has replied my comments and others properly. Accept it.

Ans) We appreciate the reviewer`s positive evaluation.

References

- 1 Seol, D. J., Lee, J. W. & Park, N. G. On the Role of Interfaces in Planar -Structured HC(NH₂)₂PbI₃ Perovskite Solar Cells. *ChemSusChem* **8**, 2414-2419 (2015).
- 2 Shao, Y. *et al.* Grain boundary dominated ion migration in polycrystalline organic–inorganic halide perovskite films. *Energy Environ. Sci.* **9**, 1752-1759 (2016).
- 3 Xiao, X. *et al.* Suppressed Ion Migration along the In-Plane Direction in Layered Perovskites. *ACS Energy Lett.* **3**, 684-688 (2018).

Reviewers' Comments:

Reviewer #1:

Remarks to the Author:

I think the authors have addressed all of my concerns. I recommend acceptance of the manuscript.

Response to Reviewer Comments

Manuscript ID: NCOMMS-18-01336

Title: 2D perovskite stabilized phase-pure formamidinium perovskite solar cells

Author(s): Jin-Wook Lee, Zhenghong Dai, Tae-Hee Han, Chungseok Choi, Sheng-Yung Chang, Sung-Joon Lee, Nicholas De Marco, Hongxiang Zhao, Pengyu Sun, Yu Huang and Yang Yang*

Department of Materials Science and Engineering and California NanoSystems Institute, University of California, Los Angeles, California 90095, United States.

First of all, we thank the referees for their valuable comments on our manuscript of MS ID: NCOMMS-18-01336 entitled “2D perovskite stabilized phase-pure formamidinium perovskite solar cells” (corresponding authors, Y. Y.). Here we have addressed the queries from the reviewers and revised the manuscript according to the reviewers’ comments.

Reviewers' comments:

Reviewer #1 (Remarks to the Author):

I think the authors have addressed all of my concerns. I recommend acceptance of the manuscript.

Ans) We appreciate the reviewer`s positive evaluation.